# Macrophage–Neuroglia Interactions in Promoting Neuronal Regeneration in Zebrafish

**DOI:** 10.3390/ijms24076483

**Published:** 2023-03-30

**Authors:** Chih-Wei Zeng

**Affiliations:** 1Department of Molecular Biology, University of Texas Southwestern Medical Center, Dallas, TX 75390, USA; chih-wei.zeng@utsouthwestern.edu; 2Hamon Center for Regenerative Science and Medicine, University of Texas Southwestern Medical Center, Dallas, TX 75390, USA

**Keywords:** TNF/Tnfrsf1a, HDAC1, macrophages, ependymal–radial glia progenitors, pMN progenitors, radial glial cells, spinal cord injury, zebrafish

## Abstract

The human nervous system exhibits limited regenerative capabilities following damage to the central nervous system (CNS), leading to a scarcity of effective treatments for nerve function recovery. In contrast, zebrafish demonstrate remarkable regenerative abilities, making them an ideal model for studying the modulation of inflammatory processes after injury. Such research holds significant translational potential to enhance our understanding of recovery from damage and disease. Macrophages play a crucial role in tissue repair and regeneration, with their subpopulations indirectly promoting axonal regeneration through developmental signals. The AP-1 signaling pathway, mediated by TNF/Tnfrsf1a, can elevate HDAC1 expression and facilitate regeneration. Furthermore, following spinal cord injury (SCI), pMN progenitors have been observed to switch between oligodendrocyte and motor neuron fates, with macrophage-secreted TNF-α potentially regulating the differentiation of ependymal–radial glia progenitors and oligodendrocytes. Radial glial cells (RGs) are also essential for CNS regeneration in zebrafish, as they perform neurogenesis and gliogenesis, with specific RG subpopulations potentially existing for the generation of neurons and oligodendrocytes. This review article underscores the critical role of macrophages and their subpopulations in tissue repair and regeneration, focusing on their secretion of TNF-α, which promotes axonal regeneration in zebrafish. We also offer insights into the molecular mechanisms underlying TNF-α’s ability to facilitate axonal regeneration and explore the potential of pMN progenitor cells and RGs following SCI in zebrafish. The review concludes with a discussion of various unresolved questions in the field, and ideas are suggested for future research. Studying innate immune cell interactions with neuroglia following injury may lead to the development of novel strategies for treating the inflammatory processes associated with regenerative medicine, which are commonly observed in injury and disease.

## 1. Introduction

A previous report described that the immune response could promote neuronal regeneration in zebrafish, especially macrophages. Macrophages are immune cells that are important for tissue repair and regeneration. In mammals, axonal repair following traumatic spinal cord injury (SCI) is dependent upon the rapid development of reparative M2-macrophages [1], because sustained recruitment of inflammatory blood-derived macrophages can facilitate extensive secondary axonal dieback and substantially delay the reparative process. However, activated macrophages can also promote axonal regeneration [2], suggesting that the immune response plays complex roles after spinal injury. In zebrafish, which contain two types of macrophages, unpolarized macrophages are recruited to the inflammation site and adopt a M1-like phenotype. Subsequently, the macrophages convert their functional phenotypes from M1-like to M2-like in response to a progressive inflammatory microenvironment [3]. Moreover, microglia are activated after SCI in adult [4] and larval [5] zebrafish, suggesting that innate immune cells have repair functions. In zebrafish, T cells can also mediate organ-specific regenerative programs, suggesting that adaptive immunity is also important for spinal cord regeneration in this species [6]. However, the mechanisms through which macrophages can promote neuronal regeneration in zebrafish, as well as the existence of a pro-regenerative macrophage subpopulation that can promote neuronal regeneration in zebrafish, are unclear.

After injury, the immune response is a significant source of non-developmental signals. Studies have demonstrated the positive effects of macrophages during regenerative neurogenesis in zebrafish. The cytokine tumor necrosis factor (TNF; also known as TNF-α) mediates a broad range of cellular activities, including proliferation, survival, differentiation, and apoptosis, and it is considered essential for inducing and maintaining the inflammatory immune response [7]. Recent studies have demonstrated that TNFα can exhibit dual roles depending on the context, with both pro-inflammatory and pro-regenerative functions. In the context of spinal cord injury, the specific subtypes of macrophages and the balance between TNFα1 and TNFα2 signaling are crucial for determining the regenerative outcome. Activated *tnfa*+ macrophages have been determined to express a mixture of M1 and M2 markers, which can facilitate axon growth and neurogenesis [8]. The interplay between TNFα1 and TNFα2 signaling is essential for modulating the switch between pro-inflammatory and pro-regenerative functions. TNFα1, typically associated with pro-inflammatory responses, can trigger cell death and impede regeneration. On the other hand, TNFα2 signaling has been linked to increased regenerative outcomes by promoting cell survival and tissue repair [9]. The activation of TNFα2 in the presence of TNFα1 can counteract the detrimental effects of TNFα1 signaling, thereby creating a favorable environment for regeneration. Furthermore, the downstream signaling pathways activated by TNFα1 and TNFα2 also contribute to their distinct cellular responses. While TNFα1 primarily activates the NF-κB pathway, leading to inflammation and cell death, TNFα2 preferentially activates the PI3K/Akt pathway, which promotes cell survival and regeneration. Understanding the delicate balance between TNFα1 and TNFα2 signaling and their distinct mechanisms in the regenerative process could provide insights for the development of targeted therapies to enhance spinal cord repair following injury.

Recent research on zebrafish has highlighted the importance of macrophages in promoting axonal regrowth. Specifically, TNF-α has been identified as a key factor in this process [10]. Inhibiting TNF-α has been shown to decrease axonal regrowth, indicating its indispensable role in the process. These findings suggest that TNF-α plays a critical role in promoting regeneration in association with other neuroglia. For instance, macrophages can indirectly affect ependymo-radial glia (ERG) progenitor cells, promoting regenerative neurogenesis in zebrafish after spinal cord injury [11]. Thus, the effects of macrophages could act indirectly by inducing the re-expression of developmental signals from the environment. Indeed, developmental signals such as Wnt, Fgf, Shh, and dopamine are re-expressed after spinal injury in zebrafish, promoting regenerative neurogenesis. For example, Wnt/b-catenin signaling is required for radial glial neurogenesis following SCI [12]. Additionally, Goldshmit et al. (2018) have demonstrated the role of Fgf in driving neural proliferation and neurite outgrowth of different spinal cord neuron populations during both neural development and adult regeneration [13]. Therefore, macrophages can affect different types of neuroglia and promote regeneration via developmental signals or non-developmental signals. 

It is now known that SCI activates macrophages with different functional phenotypes, and the activation of specific types of macrophages can result in the secretion of different cytokines that impact neuroglia and promote regeneration through non-developmental signals. Regenerative neurogenesis, which is the process of generating new neurons, is believed to rely on the interactions between macrophages and neuroglia, the support cells of the nervous system. However, despite some advances in our understanding of these interactions, the specific molecular mechanisms and the particular subtypes of macrophages and neuroglia involved in regenerative neurogenesis are not yet fully understood. In other words, there is still much to be learned about the specific roles that different macrophages and neuroglia play in the process of generating new neurons in the nervous system. Additionally, when discussing neurogenesis in the context of neuroprotection and neurite outgrowth, it is important to specify the role of neuritogenesis in these processes.

## 2. Overview of the Response of Macrophages to Spinal Cord Injury in Zebrafish

Zebrafish are a well-established model organism for studying nerve damage regeneration mechanisms, particularly after SCI. This is because of their remarkable ability to regenerate neural tissue after injury [14,15]. Macrophages, which are a type of immune cell, are crucial in the response to SCI in zebrafish. They play a vital role in the regenerative process by clearing debris and promoting tissue repair [16]. As the injury site is cleared of debris, macrophages, the primary immune cells involved in the inflammatory response, shift their phenotype from a pro-inflammatory (M1) to an anti-inflammatory (M2) state [17]. This phenotypic shift is essential for promoting tissue repair and regeneration. M2 macrophages secrete factors such as interleukin-10 (IL-10); transforming growth factor-beta (TGFβ); cytokines IL-4 and IL-10; and surface markers CD206 and CD163 [18,19]. These factors have anti-inflammatory and pro-regenerative effects, supporting the transition from inflammation to tissue repair. Below, we describe four different phases, each with distinct macrophage functions after SCI in zebrafish.

### 2.1. Phase 1: Acute Inflammatory Response

Upon SCI, zebrafish macrophages are rapidly recruited to the injury site and begin phagocytosing cellular debris. The macrophages also secrete pro-inflammatory cytokines, such as TNF-α and IL-1β [10], which attract additional immune cells to the injury site. Macrophages rapidly respond to injury-dependent changes in neuronal activity, alter their morphology to an activated amoeboid state, and remove nonfunctional neuronal cells after injury in a pro-inflammatory response. The response involves molecules such as the complement system, TNF-α, IL-1β, CXC3CL1, CX3CR1, and MHC-1 [20,21].

### 2.2. Phase 2: Resolution of Inflammation

As the injury site is cleared of debris, the macrophages shift their phenotypes to become anti-inflammatory. They secrete factors such as IL-10, TGFβ, and cytokines IL-4, IL-10, CD206, and CD163, which promote tissue repair and regeneration [22,23]. The macrophages may regulate aspects of appendage regeneration through Wnt/β-catenin signaling [24,25]. This signaling pathway affects macrophage proliferation and cytokine release, which can modulate inflammation and regeneration of the appendage.

### 2.3. Phase 3: Promotion of Axon Regeneration

Macrophages have been shown to promote axon regeneration in various models by releasing factors that enhance axonal growth, such as brain-derived neurotrophic factor (BDNF) and fibroblast growth factor (FGF) [26,27], although their specific role in zebrafish axon regeneration warrants further investigation. In addition to BDNF and FGF, zebrafish macrophages release other growth factors and cytokines that promote axon regeneration, including nerve growth factor (NGF) and insulin-like growth factor 1 (IGF-1) [28,29]. These factors act in a coordinated manner to stimulate axon growth and re-establish neuronal connections.

### 2.4. Phase 4: Modulation of Glial Scar Formation

Glial scarring is a major obstacle to neural regeneration in mammals [30]. In zebrafish, macrophages appear to play a role in modulating glial scar formation by promoting the clearance of myelin debris and releasing factors that inhibit excessive scar formation [31]. Additionally, zebrafish macrophages release factors that inhibit excessive scar formation. One such factor is chondroitin sulfate proteoglycans (CSPGs), which are known to be major contributors to the formation of a glial scar [32,33]. Macrophages in zebrafish have been shown to express enzymes that degrade CSPGs, thereby preventing the formation of an excessive glial scar.

Overall, macrophages are crucial for clearing debris, secreting pro-inflammatory cytokines, promoting tissue repair and regeneration, releasing factors that enhance axonal growth, and inhibiting excessive scar formation. The ability of macrophages to perform these functions is essential for the remarkable regenerative capacity of zebrafish after SCI. Further understanding of the mechanisms underlying this response may lead to new therapeutic approaches to treating SCI in humans.

## 3. The Role of TNF-Activated Macrophages in Pro-Regenerative Neurogenesis

The immune response plays a crucial role in the regenerative process of the spinal cord after injury [34]. Among the various immune cells involved, macrophages have been identified as key players in promoting neurogenesis [35]. Recent studies have shown that activated *tnfa*+ macrophages represent a subtype of pro-regenerative macrophages that act on spinal progenitor cells to promote axon growth and neurogenesis [11]. However, the mechanisms by which these macrophages promote regeneration are not fully understood. The TNF signaling appears to be regeneration-specific and not necessary for developmental neurogenesis; its upregulation after injury could be part of the reactivated developmental program for neurogenesis [36]. Therefore, a better understanding of the intracellular signaling that occurs in spinal progenitor cells after injury could lead to new translational approaches for spinal cord repair.

TNF is primarily produced by activated macrophages and is known to promote inflammation and cell death in certain contexts [37]. However, recent studies have shown that TNF can also play a pro-regenerative role by promoting axon growth and neurogenesis in the spinal cord [38,39]. One key subtype of pro-regenerative macrophages are the activated *tnfa*+ macrophages, which have been shown to express a mixture of M1 and M2 markers, including heparin-binding epidermal growth factor (HB-EGF), a pro-regenerative factor [40]. These macrophages can act on spinal progenitor cells to promote neurogenesis and axon growth [41,42], both directly and indirectly, through other cell types and signals in the injury site. Downstream effectors of TNF signaling include the AP-1 complex, which is involved in promoting pro-regenerative responses in salamanders [11] and has been shown to be important for TNF-induced neurogenesis in mouse SVZ neurospheres [43]. Additionally, TNF signaling upregulates key neurogenic factors such as histone deacetylase 1 (HDAC1), which is involved in the reactivated developmental program for neurogenesis after injury [44].

TNF signaling appears to be regeneration-specific and not necessary for developmental neurogenesis [10]. This may be due to the fact that TNF is primarily produced by reactive macrophages, which are only present after injury. In addition, expression of tnfrsf1a, the receptor for TNF, is dispensable for developmental neurogenesis in unlesioned animals [45]. While TNF signaling shows promise as a potential therapeutic target for promoting spinal cord regeneration, simply enhancing extracellular Tnf signaling may negatively affect other aspects of spinal repair. Therefore, a better understanding of the intracellular signaling in spinal progenitor cells after injury is needed to develop more targeted approaches for spinal cord repair.

## 4. The TNF/Tnfrsf1a Mediated AP-1 Signaling Pathway Increases Hdac1 after Injury

The TNF-mediated signaling pathway has been shown to be involved in regeneration processes. For instance, in the case of fin regeneration in zebrafish, Tnfrsf1a-mediated sensitivity to exogenous TNF is necessary [46]. This demonstrates that the TNF/Tnfrsf1a signaling pathway plays a role in promoting regeneration in certain tissues. However, it has been reported that the TNF/Tnfrsf1a signaling pathway is not crucial for regeneration in the zebrafish retina [47]. This finding suggests that the role of the TNF/Tnfrsf1a signaling pathway in promoting regeneration might be tissue- or cell-specific, rather than universally applicable across all tissues and cells. Therefore, the TNF-mediated signaling pathway is involved in regeneration, and its role appears to be dependent on the specific tissue or cell type being considered. Further research is needed to better understand the underlying mechanisms and the varying roles of the TNF/Tnfrsf1a signaling pathway in different tissues and cells. Cavone et al. (2021) reported Tnfrsf1a-mediated sensitivity to exogenous TNF in spinal progenitor cells after SCI. The TNF/Tnfrsf1a signaling pathway clearly increased fin and spinal cord cell regeneration, but the retina was not regenerated [11]. Notably, soluble TNF exerts its biological functions by binding to special target cell surface receptors, which have been identified as Tnfrsf1a and Tnfrsf1b [48]. After TNF binding, Tnfrsf1a or Tnfrsf1b can be bound by TNFR-associated protein-2 and the serine/threonine kinase receptor-interacting protein, which mediates survival and proliferating signals through the transcription factor NFκB and activation of the c-jun N-terminus kinase, which in turn mediates new gene transcription via AP-1 [49,50]. Raivich et al. (2004) discovered that the AP-1 transcription factor c-Jun plays a crucial role in axonal growth in the injured CNS. The study also revealed that reduced expression of CD44, galanin, and α7β1 integrin, molecules involved in regeneration, suggests a mechanism for c-Jun-mediated axonal growth, highlighting c-Jun’s significance in axonal regeneration in the injured central nervous system (CNS) [51]. Cavone et al. (2021) found that Tnfrsf1a-mediated exogenous TNF and activation of AP-1 are able to increase regenerative neurogenesis after SCI [11]. It is known that neuronal progenitors lacking HDAC1 and HDAC2 are unable to differentiate into mature neurons and undergo cell death [52]. In zebrafish, HDAC1 represses Notch target gene expression during neurogenesis and favors the generation of motor neurons in response to hedgehog signaling [53] to promote retinal neurogenesis [54]. These findings suggest that the activity of HDAC1 can regulate neural differentiation. TNF-α from macrophages induces Tnfrsf1a-mediated AP-1 activity in progenitors to increase regeneration-promoting expression of HDAC1 and neurogenesis [11] (Figure 1). Overall, the TNF/Tnfrsf1a-mediated AP-1 signaling pathway increases HDAC1 expression and promotes regeneration in zebrafish after injury.

Past research has shown that macrophages can support regenerative neurogenesis through developmental and non-developmental signaling pathways. They contribute to axonal regeneration through developmental signals while pro-regenerative macrophages release TNF-α, which stimulates ERG progenitors to promote regenerative neurogenesis via Tnfrsf1a-mediated AP-1 activity and Hdac1 expression, representing non-developmental signals. However, the involvement of other neuroglial cell types in this process remains unclear. In the subsequent two sections, we provide additional evidence suggesting that other neuroglia may also contribute to regenerative neurogenesis through TNF-mediated signaling pathways.

## 5. Macrophages Promote Neuronal Regeneration through ERG Progenitor Cells after Injury

It has been demonstrated that the TNF-mediated signaling pathway plays a crucial role in promoting regeneration, particularly through the activation of pro-regenerative macrophages and modulation of intracellular signaling in spinal progenitor cells after injury [2]. Additionally, microglia and the immune system play dynamic roles during successful regenerative responses in the retina [55]. Cavone et al. (2021) found that a subpopulation of macrophages directly influences the proliferation and differentiation of ependymal–radial glia (ERG) progenitor cells, promoting neurogenesis [11]. Nevertheless, the pro-regenerative effects of macrophages might also be mediated indirectly through other cell types in the spinal cord, such as through the action of TNF. Macrophages contribute to axonal regeneration by producing pro-regenerative TNF and reducing Il-1β levels [10]. These effects may lead to the suppression of Il-1β-producing neutrophils and increased TNF after injury, implying that varying levels of TNF and Il-1β could indirectly influence spinal cord regeneration. Inflammatory TNF-expressing macrophages have been found to adopt an M2-like phenotype [3]. However, Cavone et al. (2021) reported that lesion-activated TNF-expressing macrophages exhibit a mixture of M1 and M2 markers [11]. Tsarouchas et al. (2018) showed that TNF is reactive to macrophages and is present only after injury [10], indicating that lesion-activated macrophages expressing TNF represent a pro-regenerative subtype in the regeneration process.

Although it is established that the TNF-expressing macrophage subpopulation with pro-regenerative properties directly influences ERG progenitor cells for neuronal regeneration after injury, it remains uncertain whether these macrophages specifically target ERG progenitor cells or indirectly affect other cell types. Future research should investigate the selectivity of pro-regenerative macrophages towards ERG progenitor cells and explore potential direct or indirect impacts on other cell populations.

## 6. Macrophage TNF-α Signaling May Regulate pMN Progenitor Cell Fate

During vertebrate development, neural progenitor cells divide into the progenitor population and then differentiate as distinct types of neurons and glial cells. The neurons differentiate before the glial cells do. For example, pMN progenitors express Olig2, which associates with other unidentified cofactors to activate oligodendrocyte generation and repress motor neuron generation [56]. Several studies have indicated that the switch from neuron to oligodendrocyte production results from downregulation of neuronal factors and activation of glial factors [57]. Within pMN progenitors, reversible phosphorylation of Olig2 promotes the transition from motor neuron to oligodendrocyte [58]. After specification, some oligodendrocyte progenitor cells (OPCs) rapidly differentiate as myelinating oligodendrocytes [59]. On the other hand, high levels of Ngn2 favor the conversion of pMN progenitors into post-mitotic motor neurons [60] (Figure 2). However, how these various functions are integrated with the cell lineages that give rise to neurons and oligodendrocytes remains poorly understood.

Previous studies have shown that pMN progenitors can differentiate into motor neurons following SCI in zebrafish [5]. Furthermore, it has been observed that pMN progenitors switch from generating oligodendrocytes to generating motor neurons after spinal cord lesion [5], indicating that the fate of pMN progenitors can be altered based on specific molecular mechanisms in the microenvironment. For instance, the *Sox10* gene is highly expressed in OPCs and differentiating oligodendrocytes, and it encodes a transcription factor that is essential for the formation and maintenance of oligodendrocytes [61]. Additionally, the switch of motor neurons to oligodendrocyte progenitors is regulated by Prdm8, which controls the level of Shh activity in pMN progenitors, ultimately determining the myelinating fate of oligodendrocyte lineage cells [62]. Loss of Notch signaling in the pMN domain leads to increased motor neuron differentiation and a progressive depletion of pMN progenitors over time, while the activation of Notch signaling leads to a reduction in motor neurons [63]. While pMN progenitors and precursors which are closely associated with differentiating neurons express Neurog1, a pro-neuronal transcription factor, those that are more closely associated with oligodendrocyte lineage cells (as marked by the expression of Olig2 and Sox10) do not express Neurog1. These findings suggest that pMN progenitors can switch between generating oligodendrocytes and motor neurons through intrinsic signaling pathways such as Sox10, Prdm8, Shh, Neurog1, Olig2, and Notch. However, it remains unclear whether pMN progenitors can also be regulated by other cell types through extrinsic signaling pathways.

The macrophage-secreted TNF-α can promote axonal regeneration in zebrafish [10]. Recent research has demonstrated that TNF-α from pro-regenerative macrophages can activate ERG progenitors and promote neurogenesis by non-developmental signals through the expression of HDAC1 [11]. Moreover, HDAC1 is crucial for oligodendrocyte regeneration, proliferation, and nerve remyelination [64,65]. Taken together, these findings suggest that macrophage-secreted TNF-α can regulate ERG motor neuron progenitors and oligodendrocyte differentiation, ultimately promoting regenerative neurogenesis. Moreover, it has been discovered that HDAC activity is necessary for the differential repression of Nkx2.2 and Olig2 in pMN progenitors [66]. Therefore, it is possible that macrophages secrete TNF-α to regulate pMN progenitors through an extrinsic signaling pathway (Figure 3A).

Previous studies have shown that pMN progenitors can switch between oligodendrocyte and motor neuron fates through intrinsic signaling pathways, including in response to SCI. However, it is still unclear whether extrinsic signaling pathways mediated by other cell types can regulate this switch in pMN progenitors. Furthermore, while macrophages have been shown to promote neurogenesis in ERG motor neuron progenitors, it remains unknown whether macrophage-secreted TNF-α can impact pMN progenitors and drive their switch from oligodendrocyte generation to motor neuron generation after injury. Addressing these questions through future research could provide a more comprehensive understanding of the cellular mechanisms involved in regenerative neurogenesis.

## 7. Macrophage TNF-α Signaling May Regulate Radial Glial Cell-Mediated Neurogenesis

A previous study reported that radial glial cells (RGs) can promote neuronal regeneration in the CNS of zebrafish [67,68,69]. RGs in zebrafish efficiently repair lesions after injury to the telencephalon through the induction of serval processes. For example, after a lesion, the specific glial environment in the zebrafish telencephalon not only permits long-term neuronal survival, but also prevents scar formation [70]. RGs with cell bodies located at the ventricles show an increase in proliferation [71] and eventual integration of newborn and differentiated neurons [72]. Kroehne et al. (2011) also found that ventricular RG progenitor cells react to injury, proliferate, and generate neuroblasts that migrate to the lesion site by lineage tracing [73]. However, the molecular mechanisms that enable the involvement of RGs in the repair process are currently incompletely understood.

A number of studies on the spinal cord’s dorsoventral axis have established a framework for understanding the molecular mechanisms controlling neural cell diversity. These positional identity factors control the specification of both neuronal and glial subtypes. Olig2 is activated ventrally by Shh in ventricular zone precursors and underlies the sequential specification of somatic motoneurons and oligodendrocytes [74], suggesting that transcription factors control generic neurogenesis or gliogenesis. For example, neurogenesis is known to be mediated by bHLH factors [75], which are downstream targets of positional identity genes and promote cell cycle exit and pan-neuronal properties. Gliogenesis thus replaces neurogenesis, and numerous factors are known to participate in this transition [76]. For example, Sox9 is necessary for the downregulation of neurogenesis and specification of oligodendrocytes in the spinal cord [77]. The above findings suggest that RGs can perform neurogenesis and gliogenesis, which are dependent on the regulation of different transcription factors.

RGs are heterogeneously generated, and differentiate into different neuronal and oligodendrocyte cell types in the normal adult zebrafish telencephalon. RGs can differentiate into newborn cell.1 (NBN.1) or motor neurons (MNs) or NBN.2 types after injury [78]. Furthermore, the generation of OPCs from RGs occurs in the adult zebrafish brain [78]. However, whether specialized subpopulations of RGs exist for the generation of neurons and OPCs is currently unclear. A previous study showed that HDACs play crucial roles in neural development [52] and synaptic plasticity [79]. For example, HDAC1 can regulate cardiac morphogenesis and affect embryonic stem cell differentiation [80,81]. Importantly, HDAC1 also regulates RG proliferation in the developing optical tectum of *Xenopus laevis* [82], suggesting that HDAC1 acts as a positive regulator of RG proliferation in developing intact vertebrates. A previous study found that TNF-α from pro-regenerative macrophages can activate ERG progenitors and promote Hdac1 expression and neurogenesis [11]. Furthermore, TNF-α is important for oligodendrocyte regeneration and proliferation, as well as nerve remyelination [83]. However, it remains unclear which cell types or factors regulate RG to increase Hdac1 expression and promote RG proliferation.

Previous studies have demonstrated that RGs can differentiate into distinct cell types, such as NBN.1, motor neurons, NBN.2, and OPCs, depending on whether they follow the neurogenesis or gliogenesis pathway. It is also known that HDAC1 can play a role in regulating RG proliferation. However, the specific molecular mechanisms that govern the differentiation of RGs into different cell types require further investigation. Furthermore, it is unclear whether specific subpopulations of macrophages secrete TNF-α to activate RGs and promote gliogenesis. One possible pathway involves specific subpopulations of macrophages secreting TNF-α, which directly affects RGs and activates the HDAC1-mediated pathway, leading to RGs switching from gliogenesis to neurogenesis after injury. (Figure 3B). Further research could elucidate the potential role of macrophage-derived TNF-α in RG differentiation and determine whether this cytokine can promote RG differentiation toward the gliogenesis pathway.

## 8. Conclusions

In recent years, the role of immune cells in the CNS has been extensively investigated, revealing their potential to influence neuronal regeneration after injury. Macrophages, in particular, have been shown to exhibit significant plasticity and diverse functions. A subpopulation of macrophages expressing TNF and possessing pro-regenerative properties has been found to influence ERG progenitor cells, promoting neuronal regeneration after SCI in zebrafish. However, the specificity of these macrophages when targeting ERG progenitor cells, as well as their effects on other cell types, remain unclear. This review article proposes two other cell types, pMN progenitors and RGs, which may also generate neurons through TNF-α signaling and interact with macrophages to facilitate neural regeneration.

pMN progenitors, which are responsible for generating motor neurons and oligodendrocytes, could be influenced by macrophages through extrinsic signaling pathways. The pro-regenerative properties of macrophages could directly or indirectly impact pMN progenitors, promoting regenerative neurogenesis after injury. Investigating the mechanisms underlying these interactions may help us to understand the role of macrophages in neuronal regeneration and provide insights into potential therapeutic interventions. RGs are another critical cell type in the CNS, functioning as both neuronal and glial progenitors. RGs have been shown to switch from generating oligodendrocytes to motor neurons, which could be influenced by macrophage-secreted TNF-α. Furthermore, specific macrophage-secreted TNF-α can promote oligodendrocyte proliferation and nerve remyelination, which is essential for the restoration of neural function after injury.

In conclusion, this review article highlights the potential of macrophages and their subpopulations to interact with other neuroglia, such as pMN progenitors and RGs, to facilitate neural regeneration following injury. Further studies should aim to elucidate the mechanisms underlying these interactions and their implications in neuronal regeneration. Unraveling the complexity of macrophage-mediated regeneration may pave the way for the development of novel therapeutic strategies to enhance recovery after spinal cord injury and other CNS disorders.

## Figures and Tables

**Figure 1 ijms-24-06483-f001:**
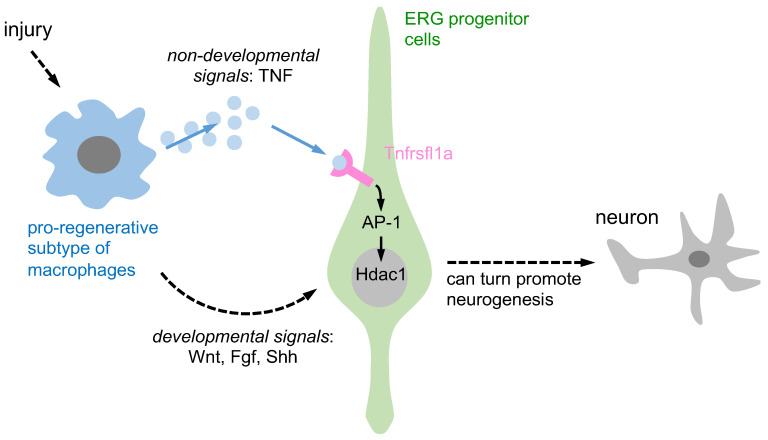
A schematic representation is provided to illustrate the activation of ERG progenitors by pro-regenerative macrophages, which in turn promote neurogenesis through two distinct signaling pathways. Pro-regenerative macrophages utilize two different signaling pathways to induce ependymo-radial glia (ERG) progenitors and promote regenerative neurogenesis. Firstly, through non-developmental signals, macrophages secrete TNF-α, which induces Tnfrsf1a-mediated AP-1 activity in ERG progenitors and promotes Hdac1 expression and neurogenesis. Secondly, through developmental signals, macrophage-secreted TNF-α can indirectly impact ERG progenitor cells by increasing developmental signals such as Wnt, Fgf, and Shh, promoting regenerative neurogenesis.

**Figure 2 ijms-24-06483-f002:**
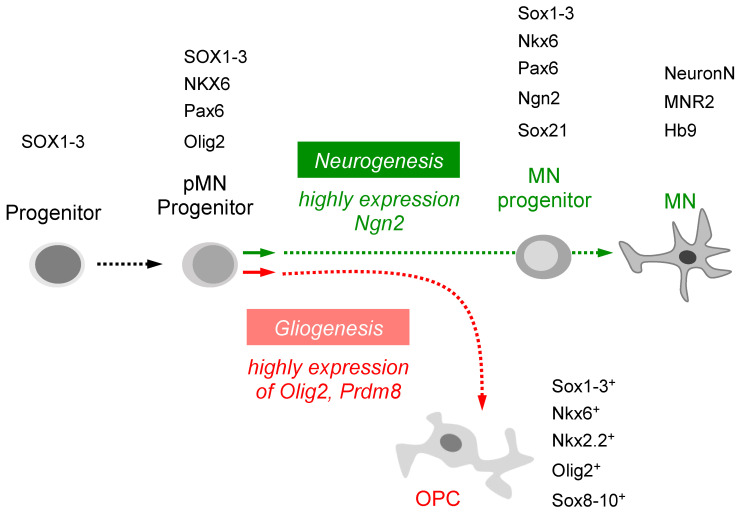
A graphic is displayed to depict the intrinsic signaling pathway through which various transcription factors can induce pMN progenitors to differentiate into distinct cell types. The specification of pluripotent stem cells and neural precursor cells is mediated by transcription factors. Neurogenesis is promoted by a high level of Ngn2, leading to the differentiation of pMN progenitors into mature motor neurons, while gliogenesis is facilitated by a high level of Olig2 or Prdm8, resulting in the differentiation of pMN progenitors into OPCs.

**Figure 3 ijms-24-06483-f003:**
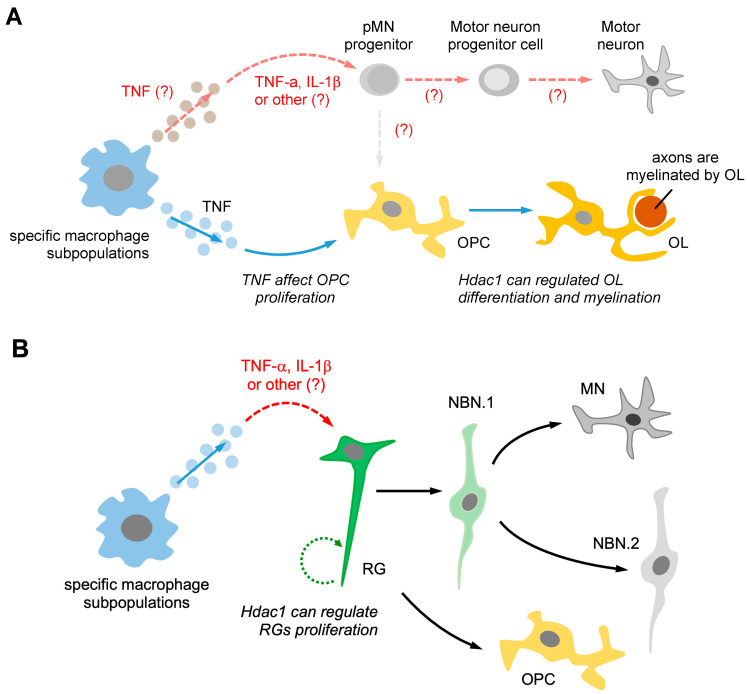
Pro-regenerative macrophages may activate different types of neuroglia, which can promote neurogenesis through an extrinsic signaling pathway. (**A**) Pro-regenerative macrophages utilize two different signaling pathways to induce pMN progenitors and oligodendrocyte progenitor cells (OPCs) to promote regenerative neurogenesis. Firstly, macrophage-secreted TNF-α may induce pMN progenitors, promoting neurogenesis instead of generating OPCs. Secondly, macrophage-secreted TNF-α can affect OPC proliferation and increase the Hdac1-mediated pathway, promoting oligodendrocyte (OL) differentiation and myelination. (**B**) The subpopulation of macrophages can secrete TNF-α, which directly induces the Hdac1-mediated pathway in radial glial cell (RG) activation, leading to RG proliferation. TNF-α secreted by macrophages may also guide RGs towards the neurogenesis pathway, promoting the generation of motor neurons (MNs) instead of NBN. The differentiation of RG into neurons is known as the neurogenesis pathway, while differentiation into OPCs is referred to as the gliogenesis pathway. The “(?)” symbol indicates questions that require further investigation in the future.

## Data Availability

Not applicable.

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
