# Peer review of "Macrophage–Neuroglia Interactions in Promoting Neuronal Regeneration in Zebrafish"

_ijms, 2023, doi:10.3390/ijms24076483_

Round 1

Reviewer 1 Report

The review "Macrophage-Neuroglia Interactions in Promoting Neuronal Regeneration in Zebrafish” describe the crucial role of macrophage and their subpopulations in tissue repair and regeneration and highlight the role of TNF-a in promoting axonal regeneration using zebrafish as animal model. The manuscript has good organization and the objectives were clearly explained and overall well written. The review is interesting but there is a need to better clarify some concepts.

Minor revisions:

Introduction: line 62-68. Please explain better this concept

Chapter 2 line 71-94: the chapter contains several repetitions (for example “however”), please rewrite this chapter and describe better the concepts you would like to underline

Chapter 3: line 96-100. The concept is not clear, please explain better

Figure Legend 1 (line 123). The entire legend should be in the same page.

Chapter 3: line 131-139. It would be better to riformulate this concept, there are some repetitions

Reviewer 2 Report

The manuscript (ijms-23077775) entitled “Macrophage–Neuroglia Interactions in Promoting Neuronal Regeneration in Zebrafish” seems good review manuscript. The manuscript discussed much about macrophage mediate axonal growth and neurogenesis in Zebrafish SCI. Usually the glial cells are resident macrophage cells in brain, hence in title mentioned both is little confusing. In this article, the authors discussed about the regenerative process on TNF, AP-1, Neurotrophic factor and HDAC1 mediated action. Still many articles suggested TNF had dual role, of which author focused much about the neurogenesis (neurite outgrowth or axonal growth), whereas inflammatory induced apoptosis and cell death didn’t discuss much. And required to discuss about the TNF directed receptor (TNFR1 & TNFR2) needs to specify the action involved through this receptor. Author discussed much neurogenesis and trafficking about the RG cells, OPC. Also, required add the neuritogenesis.

-      As the manuscript is much descriptive requires more graphical representation to explain far better.

-      Elaborate the abbreviation on initial usage.

-      Line 48-51; Promotion of neuronal regeneration in zebra fish by the macrophage states unclear but immediately sort by stating TNFa. Usually, TNFa secretion happens in neuroinflammatory mechanism leads to cell death. Hence, the author needs to explain more about the TNFa1 & 2 functioning mechanism in the regenerative mechanism in the next paragraph (DOI: https://doi.org/10.1186/s13287-020-01740-5).

-      Line 78-81; Few literatures say, on activation glia cell will leads to neurodifferentiation for the mechanism of adaptation. This mechanism enhances neuroprotection and neurite outgrowth. If any experimentally proof for neurogenesis needs to included will give advantage for manuscript. Or the neurogenesis mention in the point of neuroprotection and neurite outgrowth means needs to specify neuritogenesis.

-      Line 116-118; Check whether macrophage secretes BDNF, FGF, Confirm.

-      Line 174; Reframe the sentence more specifically.

-      Line 187-188; Instead of saying axonal regeneration can specify neurite outgrowth or axonal growth.

-      Line 285; TNFa promote axonal regeneration and neurogenesis requires to discuss about the receptor in which TNFa binding receptor and their specific action.
